

# The weather@home regional climate modelling project for Australia and New Zealand

Mitchell T. Black[1], David J. Karoly[1], Suzanne M. Rosier[2], Sam M. Dean[2], Andrew D. King[1], Neil R. Massey[3], Sarah N. Sparrow[3,4], Andy Bowery[4], David Wallom[4], Richard G. Jones[2,5], Friederike E. L. Otto[3], and Myles R. Allen[3]

[1]School of Earth Sciences and ARC Centre of Excellence for Climate System Science, The University of Melbourne, Melbourne, Australia.
[2]National Institute of Water and Atmospheric Research, Wellington, New Zealand
[3]Environmental Change Institute, Oxford University, Oxford, United Kingdom
[4]Oxford e-Research Centre, Oxford University, Oxford, United Kingdom
[5]Met Office Hadley Centre, Exeter, United Kingdom

*Correspondence to:* M.T. Black (mtblack@student.unimelb.edu.au)

**Abstract.** A new climate modelling project has been developed for regional climate simulation and the attribution of weather and climate extremes over Australia and New Zealand. The project, known as *weather@home Australia-New Zealand*, uses public volunteers' home computers to run a moderate-resolution global atmospheric model with a nested regional model over the Australasian region. By harnessing the aggregated computing power of home computers, weather@home is able

to generate an unprecedented number of simulations of possible weather under various climate scenarios. This combination of large ensemble sizes with high spatial resolution allows extreme events to be examined with more robust estimates of uncertainty. This paper provides an overview of the weather@home Australia-New Zealand project, including initial evaluation of the regional model performance. The model is seen to be capable of resolving many climate features that are important for the Australian and New Zealand regions, including the influence of El Niño-Southern Oscillation on driving natural climate

variability. To date, 75 model simulations of the observed climate have been successfully integrated over the period 1985–2014 in a time-slice manner. In addition, multi-thousand member ensembles have also been generated for the years 2013, 2014 and 2015 under climate scenarios with and without the effect of human influences. All data generated by the project is freely available to the broader research community.

## 1 Introduction

Extreme weather and climate-related events often have a serious impact on our economy, environment and society. This is particularly true in Australia and New Zealand where recurring heatwaves, floods, droughts and wildfires have resulted in the loss of life, property and livelihoods. In the aftermath of these events the scientific community is often faced with the task of quantifying the link to different causal factors, including human-induced climate change. The delivery of such information in a timely, clear and reliable manner is an ongoing challenge. Therefore, developing a capacity to research extremes and

understand their causes continues to be crucial for predicting and managing their impacts.



There is clear evidence that the climate has changed as a result of human influence (Stocker et al., 2014) and that some aspects of extremes have changed across the globe as a result (Seneviratne et al., 2012). However, this does not imply that the occurrence of every recently observed extreme weather or climate-related event was the result of human influence on the climate system, as such events may still have occurred (however unlikely) in the absence of such an influence (e.g., Stott et al.,

2013). This can be understood by recognising that our climate is a complex, chaotic system that is influenced by internal and external forcings. Processes that generate internal climate variability include atmosphere-ocean teleconnections, such as El Niño-Southern Oscillation (ENSO), as well as chaotic natural variability. Meanwhile, external forcings of climate can be either natural, such as explosive volcanic eruptions, or anthropogenic, such as greenhouse gas emissions from the burning of fossil fuels.

Distinguishing between internal and external climate forcings becomes increasingly difficult when moving from global to regional scales, due to a lower signal-to-noise ratio (Karoly and Wu, 2005; Stott et al., 2011). That is to say, the noise associated with internal climate variability is greater at regional scales than at the global scale. This is particularly true for Australia, which is recognised as having one of the most variable climates in the world (e.g., Nicholls et al., 1997). The variability in Australia's climate is driven by a number of factors, in particular the year-to-year variations in sea surface temperatures in both the Pacific

and Indian Oceans (Risbey et al., 2009). ENSO represents the variations in sea-surface temperatures and atmospheric patterns across the Pacific Ocean, with warm (El Niño) conditions producing below-average rainfall, above-average temperatures and often drought over much of northern and eastern Australia (McBride and Nicholls, 1983; Holland, 1986; Jones and Trewin, 2000). The reverse is true during cool (La Niña) conditions. Although New Zealand's climate is not usually affected as strongly by ENSO as are parts of Australia, there is nevertheless a significant influence (e.g., Gordon, 1986). In addition to ENSO there

are a number of other drivers of natural climate variability for Australia and New Zealand, including the Southern Annular Mode (Hendon et al., 2007), variations in storm tracks (Frederiksen and Frederiksen, 2007) and atmospheric blocking (Pook and Gibson, 1999). Therefore, any assessment of extreme weather and climate-related events needs to consider the interplay of both natural climate variability and forced external changes, such as the warming effect caused by increased greenhouse gas emissions.

In light of this challenge, an emerging field of climate science (known as event attribution) is seeking to quantify how the risk of weather and climate-related extremes has changed as a consequence of particular forcings acting on the climate system (Allen, 2003; Stott et al., 2004; Pall et al., 2011; National Academies of Science, Engineering and Medicine, 2016). This is typically achieved by comparing the probability of such events under the current (observed) climate against that for counterfactual worlds in which particular forcing factors (such as human-induced climate change) are absent. We of course are

unable to observe a world in which either anthropogenic or natural forcing is absent, therefore physically based climate models are required to estimate how the climate would respond to anthropogenic and natural forcings (Hegerl and Zwiers, 2011).

Undertaking event attribution studies of extreme weather events is typically restricted by two important modelling requirements: ensemble size and model resolution. Extreme weather events are, by definition, rare and therefore very large ensembles of climate model simulations are needed in order to study the event with a high degree of confidence. Meanwhile, as many

extreme events occur at a regional or local scale, the model must have sufficient resolution to realistically capture the event.



Due to these requirements, such an undertaking would be computationally expensive and typically beyond the capability of conventional computing resources. However, these demands may be met through the aggregated power of distributed computing projects. Proposed by Allen (1999) and launched in 2003, climate*prediction*.net became the largest climate modelling experiment to date by running climate models on volunteers' home computers. While the project was originally focussed on

running low-resolution global coupled atmosphere-ocean (Frame et al., 2009) and medium-resolution atmosphere-only models (Pall et al., 2011), a recent advancement (known as weather@home) involves running the global atmosphere-only model with a nested higher-resolution regional model to generate very large ensembles of model simulations (Massey et al., 2014). This regional model configuration has been implemented and evaluated over Europe (Massey et al., 2014) and the Western United States (Li et al., 2015; Mote et al., 2015) and successfully used in a number of event attribution studies (e.g., Peterson et al.,

2013; Herring et al., 2014, 2015; Schaller et al., 2016). Given the success of these existing weather@home regional climate modelling projects it was decided to implement a regional configuration over Australia and New Zealand.

The primary purpose of this paper is to provide a description and basic evaluation of the weather@home Australia-New Zealand modelling setup. This is achieved by comparing the regional model output with observations from the recent past over regions of Australia and New Zealand. Because the modelling setup is intended to be used for event attribution studies,

particular focus is given to an assessment of how well the model represents 1) mean spatial fields and interannual climate variability, 2) regional teleconnections to ENSO and 3) the distribution of daily variables at regional and local scales. For the purpose of this study we have restricted our analysis to only consider temperature and precipitation as these are the variables most commonly assessed in event attribution studies.

The remainder of this paper is structured as follows: Section 2 describes the model setup and summarises the experimental

design for the representation of the observed climate, while Section 3 provides details on the evaluation of the system. As a thorough comparison with observations is beyond the scope of this paper, we provide some illustrative comparisons of both temporal and spatial patterns. Section 4 describes how the counterfactual climate scenarios are constructed for the purpose of undertaking event attribution studies. The main conclusions are given in Section 5, including plans for future improvements.

## 2   Model description

Weather@home Australia-New Zealand uses the the Hadley Centre Atmospheric General Circulation Model 3P (HadAM3P; Massey et al., 2014) with an embedded regional model (HadRM3P; Jones et al., 2004) over the Australasian CORDEX domain (Figure 1). The HadAM3P/HadRM3P model formulation is based on the atmospheric component of the HadCM3 general circulation model (Gordon et al., 2000) with a number of improvements with respect to the calculation of clouds and convection, and a more realistic coupling of vegetated surfaces with the soil (Massey et al., 2014). HadAM3P/HadRM3P is a grid-point

model which solves equations of motion, radiative transfer and dynamics explicitly on the same scale as the grid. HadAM3P is integrated with a 15 minute timestep, has 19 vertical levels and has a regular latitude-longitude grid (1.25° longitude by 1.875° latitude) with regular poles. HadRM3P has a 5 minute timestep, has 19 vertical levels and uses a rotated grid (0.44° longitude by 0.44° latitude) with an artificial north pole at 60.31° N, 141.38° E for the Australia-New Zealand configuration.



This allows the region of interest to lie about the equator of the rotated grid, thus ensuring that each grid box in the nested region has approximately the same area. HadAM3P/HadRM3P are run in an interleaved manner: HadAM3P first runs for a full model day, providing the lateral boundary conditions to HadRM3P, which then also runs for one full model day. The coupling is strictly one-way, meaning that there is no feedback from the regional model to the global model. There is a four-point buffer

zone around the perimeter of the regional model, where the lateral boundary conditions are relaxed to values temporarily interpolated from 6-hourly output from HadAM3P. Further details of the HadAM3P/HadRM3P configuration are provided by Massey et al. (2014), but with the European region replacing the Australasian region.

Weather@home is able to generate very large ensembles of climate model simulations by harnessing spare CPU time on a network of volunteers' personal computers. This distributed computing capacity is made possible by the Berkeley Open

Infrastructure for Network Computing (BOINC; Anderson, 2004) open source infrastructure. Each volunteer signs up for the weather@home project via the BOINC client software, which automatically downloads the climate model setup to the volunteer's computer. Individual workunits are then received from the BOINC server and run when the computer is idle. The workunit contains all necessary configuration inputs needed by the climate model to run the experiment for one model year (December–November), under a specific climate scenario. After the completion of each model month the output is post-

processed to retain only a selection of key meteorological variables. This is required in order to minimise file size for data transfer and storage. A complete listing of these output variables is provided as supplementary material. Following this post-processing stage the final results are returned to a server hosted at the Tasmanian Partnership for Advanced Computing in Hobart, Australia. On average, it takes a standard home computer around 4–5 days to integrate over the model year. At the completion of the model year an additional file (the restart file) is returned that represents the final state of the atmosphere.

This final state can then be incorporated as the initial conditions for a new workunit describing the next year of the climate scenario. Therefore, this allows the system to run for a year at a time, in a time-slice manner, to generate an extended timeseries of climate model integrations. As this resubmission process is not automated, the generation of these continuous model runs is somewhat restricted by the need for project scientists to manage restart files and work unit regeneration. Therefore, typical event attribution studies will only generate multi-thousand ensemble members for a single model year of interest.

In order to represent the range of internal variability that is possible with the model, a perturbation is applied to the initial conditions of each workunit. These perturbations are applied to the global climate model in the form of slight changes to the three-dimensional potential temperature field. The initial condition perturbations were generated by calculating the next-day differences within a 1-year integration of the global model and then multiplying by five global scaling factors (1.1, 1.2, 1.3, 1.4 and 1.6) (see Massey et al. (2014) for details). This resulted in the generation of 1740 different initial condition perturbations.

In the case of an extended (multi-year) model experiment, these perturbations are only applied to the initial condition of the first model year; no perturbation is applied thereafter so as to allow for the continuous integration of the model under its specific climate scenario. Further initial condition perturbations are also applied to the first year of the model integration by selecting a range of starting conditions with different large-scale circulations and soil moisture amounts. Although all of the initial condition perturbations are only applied to the global model, they immediately affect the regional simulations through

the previously described transfer of lateral boundary conditions at the end of the first global model day.



As HadAM3P and HadRM3P are both atmosphere-only models, they require specified forcings at the boundary between the atmosphere and ocean. These lower boundary conditions come in the form of prescribed sea surface temperature (SST) and sea ice fraction (SIF) fields. For the historical 1985–2014 climate scenario used in this paper for the purpose of model evaluation, both the SST and SIF fields were sourced from the UK Met Office Operational Sea Surface Temperature and Sea Ice Analysis (OSTIA) dataset (Donlon et al., 2012). OSTIA provides global, daily fields with a spatial resolution of $0.05°$ latitude $\times$ $0.05°$ longitude. In order for these fields to be defined per grid box of the global climate model, they are regridded to the HadAM3P resolution of $1.875°$ latitude $\times$ $1.25°$ longitude using an area-weighted averaging method. Any discrepancy between the HadAM3P and OSTIA land-sea masks is resolved by taking the mean of surrounding ocean grid points. In addition to these lower boundary conditions, the model also requires the atmospheric composition to be specified. The concentrations of greenhouse gases ($CO_2$, $CH_4$ and $N_2O$), ozone, halocarbons, sulphur species and solar anomalies are all prescribed to follow the recommendations outlined by the Coupled Model Intercomparison Project Phase 5 (CMIP5; Taylor et al., 2012). An overview of the model boundary conditions for the counterfactual climate scenarios is presented in Section 4.

## 3 Model evaluation

In order to establish how well weather@home represents the observed climate over Australia and New Zealand, we generate 75 model simulations for each year over the period December 1985 to November 2014. While the modelling setup is capable of generating much larger ensemble sizes, this would be unnecessary for characterising the climatology and overall distribution of climate variables over a 29-year period. Therefore, computing resources were directed towards generating very large ensembles of climate simulations for the individual years of 2013, 2014 and 2015, for subsequent use in event attribution studies that are beyond the scope of the current paper.

The spatial fields of the regional model output are separated into six regions (Figure 1) for subsequent examination. Australia has been broken up into five established regions based on distinct climatic zones: northern Australia (NAUS), central Australia (CAUS), eastern Australia (EAUS), southwest Australia (SWAUS) and southeast Australia (SEAUS) (see CSIRO and Bureau of Meteorology (2015) for details). Meanwhile, the North and South Islands of New Zealand (NZ) are treated as a single region. While each of the six regions could have been further broken down for sub-regional detail, it would be impractical to present such a mass of information here.

### 3.1 Observational datasets

Evaluation of weather@home is undertaken by comparing the regional model output to two observational data sets: for evaluation over Australia we use the Australian Water and Availability Project dataset (AWAP; Jones et al., 2009) and for New Zealand we use the Virtual Climate Station Network dataset (VCSN; Tait et al., 2006). The AWAP dataset provides daily and monthly gridded fields of rainfall and temperature extending back to 1911 on a $0.05° \times 0.05°$ grid, and is highly regarded for studying trends and variability over Australia (e.g., Risbey et al., 2013; Min et al., 2013; King et al., 2013; Perkins and Alexander, 2013). For the purpose of this study we have masked the AWAP data over inland regions of Australia where there is





low station density. Analogous to AWAP, VCSN provides high resolution ($0.05°$ latitude $\times$ $0.05°$ longitude) estimates of daily rainfall and temperature over New Zealand extending back to 1972.

In order to compare model output and observations, a remapping of the observational datasets onto the HadRM3P model grid is required. For temperature this is achieved using bilinear interpolation. For precipitation, a conservative remapping scheme

is used to ensure that the total amount of precipitation in the remapped data is the same as in the original data.

### 3.2 Climatological mean fields and inter-annual variability

We first examine weather@home's ability to correctly represent seasonal mean fields of temperature and precipitation. By averaging over the 29-year period we are attempting to reduce the internal atmospheric model variability about the mean state. Therefore, any differences between the observations and model output may be interpreted as model deficiencies. We have

separated our analysis into seasons and for brevity are only showing results for the Austral summer (December–February) and winter (June–August).

The spatial fields of seasonal average maximum temperature (Tmax) and minimum temperature (Tmin) are shown in Figure 2 and Figure 3, respectively. In each of these figures, the top panels show the mean field from HadRM3P, averaged over the 75 ensemble members for the 29-year period, while the middle panels show the mean for the observational datasets. The

bottom panels show the difference between the model and observations and can be interpreted as an indication of model bias. Overall, the model is able to capture the large-scale spatial patterns of temperature very well, including the regions of warmest temperature over northern parts of Australia and the persistently cooler temperatures over southeast Australia and New Zealand, associated with topography. For Tmax, HadRM3P is capable of representing mean summertime values to within $\pm$ 1° C over most parts of Australia and over the North Island of New Zealand (Figure 2e). In winter, the model underestimates

Tmax at almost every land grid point across the model domain (Figure 2f). For Tmin, the model overestimates summertime values at most locations, particularly over southwest and southeast Australia and the northern and eastern parts of New Zealand (Figure 3e). In winter, the model overestimates Tmin in the north and east of Australia and in parts of New Zealand, while it underestimates temperatures to the west (Figure 3f). The prominent negative bias in temperatures along the western coastline of the South Island of New Zealand in both seasons may be the result of two features: an inability of the model to correctly

resolve temperature in this region of complex topography, as well as possible limitations in the VCSN network due to a lack of stations at high elevations.

The simulated patterns of seasonal average precipitation (Figure 4) clearly demonstrate weather@home's ability to capture both seasonal variations and, at least to some extent, the influence of topography. Over Australia, the regional model is able to capture the strong summertime monsoon rainfall over the northern parts of the continent, as well as the rainfall associated with

onshore moisture transport along the eastern seaboard (Figure 4a). There is a distinctly different rainfall distribution over Australia in winter, with the highest rainfall restricted to the southern parts of the continent, including regions of topography, where rainfall is often associated with the passage of frontal systems (Figure 4b). Over New Zealand, the model is able to resolve the strong rainfall gradient along the South Island, reflecting the region's complex topography. Overall, weather@home tends to underestimate rainfall in both seasons over Australia and New Zealand, with the exception of parts of southwest and eastern



Australia in summer (Figure 4e), and parts of the South Island of New Zealand in winter (Figure 4f). The prominent differences in wintertime rainfall along the southern and western coastlines of Australia (model $\leq 50\%$ of observations; Figure 4f) suggest that the model may not be able to fully capture the influence of local land sea breezes and/or the influence of passing frontal systems at those locations. Meanwhile, the prominent differences over northern Australia in JJA (Figure 4f) are an artefact of

expressing the differences as percentages; the actual rainfall values for both observations and the model output are both small over this region, meaning that any resulting small differences equates to a large percentage.

Next, we assess HadRM3P's capacity to represent interannual variability. For each of the regional clusters identified in Figure 1, we compare timeseries of annual variations of seasonal average temperature and precipitation from the model and the observational datasets (Figures 5–7). Here, we express these timeseries as anomalies relative to the period mean. The solid

lines show the median of the 75 ensemble members, while the shading represents the 5th–95th percentile range. Meanwhile, the dashed line represents the corresponding observational dataset. For brevity, we only show timeseries for summer here (winter timeseries are included as supplementary material).

There is general agreement between the interannual variability captured by weather@home and the observational datasets (Figures 5–7). For each of Tmax, Tmin and precipitation there are individual years that correspond to peaks/troughs in both

the model estimates and observations, and the overall shapes of the curves are similar. In addition, the observations lie within the model ensemble range for each year with only a few exceptions (e.g., northern Australian rainfall; Figure 7a) . Because sea surface temperatures are the only source of interannual variability that is common to both the weather@home simulations and the observational records, the agreement between the timeseries (as represented by the correlation coefficients) in Figures 5–7 highlights the importance of sea surface temperatures on driving the climates of Australia and New Zealand.

## 3.3    Response to ENSO

Given that ENSO is an important driver of natural climate variability for Australia and New Zealand (e.g., Nicholls et al., 1997), we assess weather@home's ability to correctly simulate ENSO teleconnections. This is achieved by comparing regional model output against observations for the different phases of ENSO: La Niña, neutral and El Niño. La Niña (El Niño) events were defined when the average Nino-3.4 index was at or below (above) $-1°$ C ($+1°$ C) anomaly for at least three months in the

September–February period. Neutral events were defined as periods when the average Nino-3.4 index did not go beyond $\pm 1°$ C anomaly in any month of September–February. These criteria allowed an equal number of events to be selected for each of the three ENSO phases: La Nina (1988/1989, 1998/1999, 2007/2008, 2010/2011), neutral (1992/1993, 1993/1994, 2003/2004, 2012/2013) and El Niño (1994/1995, 1997/1998, 2002/2003, 2009/2010). Furthermore, these criteria allowed the events to be relatively evenly spread across the period of available model simulations (1985–2014). The events were grouped according to

their ENSO phase for subsequent analysis.

Figures 8–10 show the distributions of temperature and rainfall, averaged over September–February, for the different phases of ENSO. For the purpose of model evaluation we present results for each of the six study regions. The model-derived distributions are shown as box and whisker plots; each box represents the median and first and third quartiles, while the whiskers extend to the 5th and 95th percentiles. Meanwhile, the corresponding values calculated from the observational dataset are rep-





resented as dots. It is worth noting that while we only have four examples of observed atmospheric response to each of the La Niña, neutral and El Niño forcings, we have $4 \times 75$ examples from weather@home. This large number of model simulations allows us to reduce the influence of internal chaotic variability by averaging in the modelled ensemble and thus, differences between the median values of the box plots are likely to be representative of forced responses to the observed teleconnections.

Overall, weather@home is able to correctly represent the response of temperature and rainfall to changes in the phase of ENSO. As ENSO changes from the La Niña to El Niño phase, there is a warming shift in the distributions of Tmax over each of the Australian regions (Figure 8a–e). Meanwhile, the model is able to capture the reverse relationship for Tmax over New Zealand (Figure 8f). The response of Tmin to changes in the phase of ENSO is less pronounced over Australia (Figure 9a–e), while conditions continue to be warmer over New Zealand during the La Niña phase (Figure 9f). For precipitation, there is

a shift towards higher rainfall totals over each of the study regions during La Niña conditions (Figure 10). Weather@home seems able to capture the observed non-linear ENSO-precipitation relationship despite many global coupled and atmosphere-only models failing to do so (King et al., 2015b). While the limited observations prevent us from determining if the magnitude of these shifts are suitably represented by the model, they do suggest that the directions of these shifts are correct.

## 3.4 Daily variability

Because the weather@home setup is specifically designed for use in the attribution of extreme weather events, it is important that the model is able to correctly represent the distribution of daily values of temperature and precipitation at regional and local scales. Such an assessment needs to consider not only the model's ability to correctly resolve the mean state, but also the tails of the distributions where the extreme events lie. By identifying any limitations of the model, a bias correction approach may be used to correct for systematic errors.

Here, we compare the distributions of daily Tmax, Tmin and precipitation from the ensemble of weather@home regional simulations against the distribution of these variables in the observational dataset. By way of example, results for the SEAUS region are presented in Figure 11 for both summer (DJF) and winter (JJA) in the form of quantile-quantile plots. The corresponding plots for the other regions are presented as supplementary material. For brevity, we do not intend to provide a thorough assessment of the model's representation of daily fields over each of the defined study domains. Rather, we highlight

how the large ensemble provided by weather@home allows us to systematically identify biases in the modelled distribution.

  Figure 11 is constructed by extracting the daily fields of Tmax, Tmin and precipitation from the regional model simulations and calculating area averages over the SEAUS region. As the models uses a 30-day calendar, and there are 2175 model simulations (75 model realisations for each of the 29 years), this results in a total sample size of 195,750 daily values for each season. This large sample size provides a thorough sampling of physically plausible climate states represented by the model and thus, allows the tails of the distribution to be resolved with confidence. The solid blue line in Figure 11 identifies

the percentile values when considering all of the 2175 model runs together, while the envelope shows the 5th to 95th percentile range for values at each percentile when considering each model run separately. Therefore, the range of this envelope provides an assessment of both model uncertainty and internal variability.



Figure 11a shows that the weather@home regional model provides an almost perfect representation of the distribution of daily summertime Tmax averaged over SEAUS, when compared against the AWAP observational dataset. That is to say, the solid blue line is almost directly overlying the 1:1 line of agreement (shown in black). The model is capable of not only correctly resolving the mean state, but also the tails of the distribution (represented by the 1st and 99th percentiles). During winter, the

model is seen to underestimate daily Tmax over SEAUS (Figure 11b). This bias increases when moving from the lower to higher percentiles, and the spread in values (represented by the envelope) becomes larger and asymmetrical. This large spread in the envelope indicates that the model is capable of representing a wide range of temperatures and therefore, in order to fully sample internal variability of the model, a large ensemble is necessary. For Tmin, there is a fairly consistent positive bias during summer (Figure 11c), while in winter this positive bias is really only evident at the higher percentiles (Figure 11d). The regional

model appears to consistently underpredict precipitation over SEAUS in both summer (Figure 11e) and winter (Figure 11f). In the case of the model bias being fairly consistent, a straightforward scaling and offset bias correction approach could be used. For cases when there is more inconsistency in the model (e.g., underpredicting the lower percentiles and overpredicting the higher percentiles) a more complicated bias correction technique could be employed, such as quantile-matching.

The relatively high resolution of the weather@home regional model, and the large number of model simulations, allows the

performance of the model to also be assessed at much more local scales. By way of example, quantile-quantile plots have also been generated for the coastal city of Melbourne and the inland city of Mildura by extracting and examining the corresponding nearest model grid point (see supplementary material). Overall, the weather@home setup provides sufficient model resolution and ensemble sizes to allow the model to be assessed (and where appropriate, bias corrected) for subsequent use in event attribution at both the regional and local scales.

**4   Creating counterfactual climate scenarios**

In order to quantify how human-induced climate change has altered the likelihood of extreme weather and climate related events, large ensembles of model simulations are required under two distinct scenarios: under current (observed) climate and under a counterfactual (natural) climate as might have been without human influence on atmospheric composition. Up to this point we have only considered the weather@home model under the observed climate scenario. Therefore, a brief description

of the counterfactual climate scenarios is provided here.

The key differences between the observed and counterfactual climate scenarios are the boundary conditions used to drive the weather@home model. As outlined in Section 2, simulations for the observed climate are driven by observed SSTs and sea-ice from the OSTIA dataset, as well as present day atmospheric composition (well-mixed greenhouse gases, ozone and aerosols). For the counterfactual climate the model is driven by different atmospheric composition and different sea ice and

SST specifications; the atmosphere has prescribed pre-industrial levels of greenhouse gases, ozone and aerosols, the sea ice extent corresponds to the year of maximum sea ice extent in each hemisphere of the OSTIA record, and SSTs are modified to remove estimates of anthropogenic warming. Meanwhile, boundary conditions common to both scenarios are the natural forcing factors, such as changes in volcanic aerosols and solar irradiance.



As the true climate conditions for the 'world without humans' cannot be known, weather@home simulations are run under ten alternative realisations of the counterfactual climate scenario. These alternative realisations are derived from different estimates of the underlying SST warming (delta-SST) due to human influence, which are separately calculated from ten available Coupled Model Intercomparison Project Phase 5 (CMIP5) models (Taylor et al. 2012; see supplementary material for details).

Monthly-average delta-SST estimates are calculated for each of the CMIP5 models by calculating the difference between the decadal-average (1996–2005) SSTs from the "historical" simulations (which include both anthropogenic and natural forcings) and the corresponding "natural" simulations. The resulting patterns and magnitudes of warming are seen to differ across the ten delta-SST estimates (Figure 12). These delta-SST patterns are then subtracted from the observed OSTIA SSTs to provide the lower boundary conditions for each of the respective counterfactual realisations.

The use of multiple realisations of the counterfactual scenario allows us to account for some of the uncertainty in our estimates of a world without anthropogenic influence. By way of example, Figure 13 shows weather@home model estimates of summertime daily Tmax at Melbourne for 2014/2015, under the observed and counterfactual climate scenarios. For each of these scenarios the model has been run thousands of times and the daily values of Tmax have been extracted from the nearest model grid point to Melbourne. The return time curve for the observed climate scenario (shown in red) is positioned to the left

of the curves for the respective counterfactual scenarios (shown in shades of grey). This suggests that anthropogenic climate change has shifted the distribution of Melbourne summertime maximum temperatures towards warmer conditions. However, the extent of this shift varies when considering each of the separate counterfactual scenarios. The multiple realisations of the counterfactual climate scenario allow for uncertainty to be quantified and communicated in any resulting attribution statement. More detailed examples of event attribution studies performed using the weather@home Australia-New Zealand system can be

found in the 2015 special issue of the Bulletin of the American Meteorological Society investigating extreme events of 2014 (e.g., Black et al., 2015; Grose et al., 2015; King et al., 2015a; Rosier et al., 2015). The model evaluation undertaken in each of these studies was tailored to the region of interest and builds upon the general model evaluation in this paper.

## 5  Discussion and conclusions

The weather@home Australia-New Zealand climate modelling setup has been described and briefly evaluated. By harnessing

spare computing power of volunteers' home computers, weather@home is capable of generating very large ensembles of regional climate model simulations over Australia and New Zealand. This provides a unique tool for undertaking attribution studies of extreme weather and climate events in the region. To date, 75 model simulations have been successfully integrated over the period 1985–2014 in a time-slice manner, while multi-thousand member ensembles have also been generated under both observed and counterfactual climate scenarios for the years 2013, 2014 and 2015. All of this model output is freely

available to the research community.

The weather@home regional model is seen to be capable of resolving many climate features that are important for the Australia and New Zealand regions. This is reflected in the model's ability to provide a good representation of temperature and precipitation, both spatially and temporally. Results presented here suggest that the model is capable of correctly simulating





ENSO teleconnections, which is a key requirement given the importance of ENSO on driving natural climate variability in the region. The reasonably high resolution of the regional model, and the large ensemble size achieved through the distributed computing setup, allows extreme events to be examined with confidence at regional and local scales. While the model is seen to exhibit varying degrees of bias in temperature and precipitation for different regions, this bias may be corrected through a simple scaling and offset approach (e.g., Sippel and Otto, 2014; Mera et al., 2015), or through more complicated approaches such as quantile mapping (e.g., Bergaoui et al., 2015) or ensemble re-sampling techniques (e.g., Sippel et al., 2015).

Despite the strengths of weather@home, it is important to recognise some of the limitations of the project. Because the modelling setup relies on the aggregated power of volunteers' home computers, restriction on machine CPU means that the regional modelling setup can only be achieved using atmosphere-only models. Therefore, climate realisations using high-resolution coupled ocean-atmosphere models are not possible at this point in time. Furthermore, weather@home only uses a single atmospheric model, meaning that any resulting attribution statement can only be made within the context of that specific modelling setup. In order to test the dependence of the model simulation on physical parameterization, future work will employ a perturbed physics approach whereby perturbations will be applied to components of atmospheric and surface physics.

Other areas of current and future work involve generating larger ensembles for the recent past (1985–2015), as well as sets of future simulations under varying projections of climate change. In addition, the model will be driven with idealised SSTs for different phases of ENSO, under both current and counterfactual climate scenarios, so as to provide a novel framework for assessing the relative roles of ENSO and anthropogenic climate change on recent extreme weather events. Overall, the weather@home Australia-New Zealand modelling setup provides a unique modelling resource and greatly enhances Australia and New Zealand's capacity for researching extremes and understanding their causes.

## 6 Code availability

The HadAM3P and HadRM3P models are both available from the UK Met Office as part of the Providing REgional Climates for Impacts Studies (PRECIS) program. Access to standard versions of the software is dependent on attendance at a PRECIS training workshop after which all source and other materials is made available (http://www.metoffice.gov.uk/research/applied/applied-climate/precis/obtain). As a program for supporting developing countries this workshop is free for officially catagorised developing countries and attracts a charge for other country participants. The code to manage and embed these models within the Weather@Home project is specific to their utilisation within the BOINC environment and we consider not within the scope of this publication.

*Author contributions.* Mitchell Black's contribution towards this work was performed as part of his PhD project. The weather@home Australia-New Zealand project was initiated by Myles Allen and David Karoly and was set up with the assistance of Richard Jones, Neil Massey, Andy Bowery, Mitchell Black, Suzanne Rosier, Sam Dean, Sarah Sparrow and Friederike Otto. All results were plotted and analysed by Mitchell Black with advice from David Karoly and Andrew King. The paper was written in its final form by Mitchell Black with input from all contributing authors.



*Acknowledgements.* M. T. Black, D. J. Karoly and A. D. King have been supported by funding from the ARC Centre of Excellence for
Climate System Science (Grant CE110001028). Weather@home ANZ is a collaboration among the University of Oxford, the UK Met
Office, the ARC Centre of Excellence for Climate System Science in Australia, NIWA in New Zealand, the University of Melbourne, the
University of Tasmania and the Tasmanian Partnership for Advanced Computing. We thank the volunteers who donated their computing time
5   to run weather@home.



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







**Figure 1.** Domain and elevation of terrain (metres) used in the weather@home regional model simulations. Land areas have been separated into six regions for subsequent evaluation: northern Australia (NAUS), central Australia (CAUS), eastern Australia (EAUS), southwest Australia (SWAUS), southeast Australia (SEAUS) and New Zealand (NZ). The coastal city of Melbourne and the inland city of Mildura are identified by the asterisk symbols.





**Figure 2.** Seasonal average maximum temperatures (° C) for 1985–2014 for DJF (left), and JJA (right), from the weather@home regional model HadRM3P (top), and the observational datasets (middle; AWAP over Australia and VCSN over New Zealand). The bottom panels show the difference between HadRM3P and the corresponding observational dataset.





**Figure 3.** As in Figure 2, but showing seasonal average minimum temperature (° C).





**Figure 4.** As in Figure 2, but showing seasonal average precipitation (mm/day). The difference fields are expressed as percentages relative to the observational datasets.



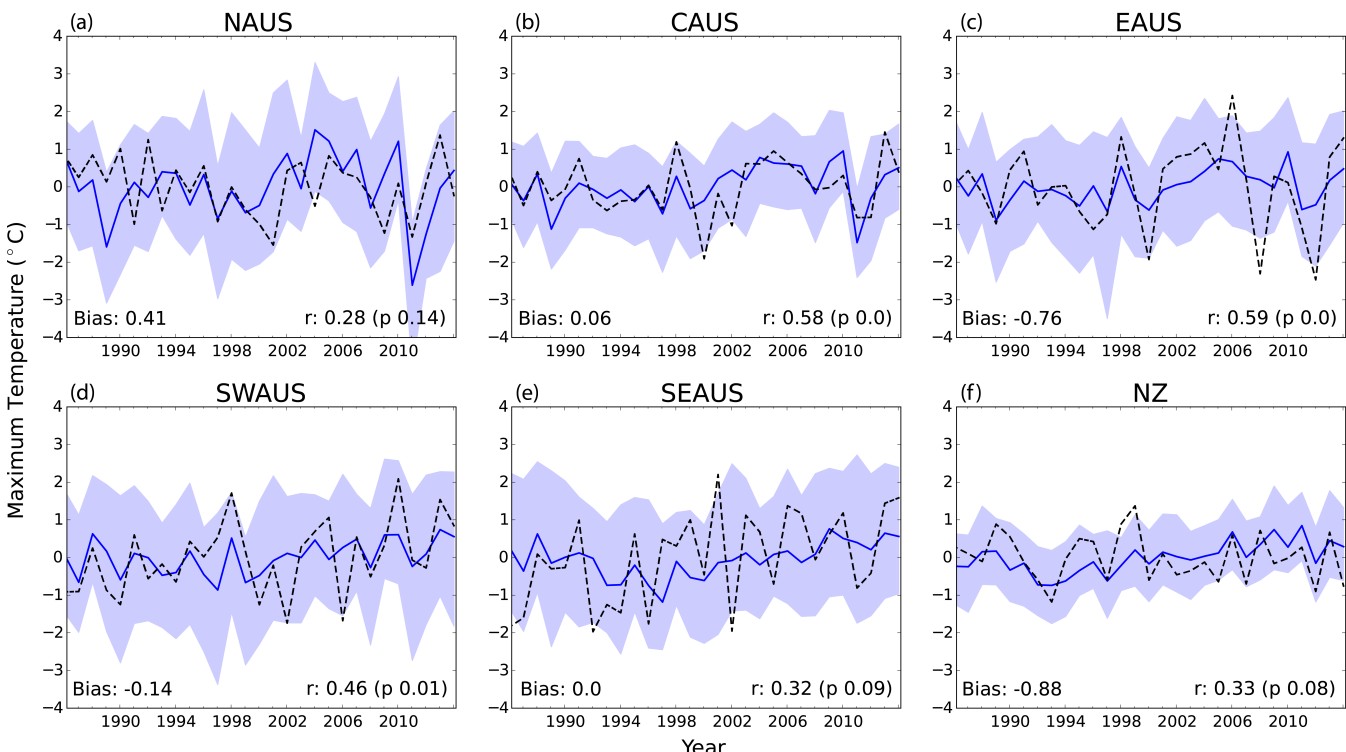

**Figure 5.** Time series of summertime (December–February) average maximum temperature for the respective study regions (as labelled) for 1986–2014. Ensemble-mean values from weather@home simulations are shown by the solid line (5th–95th percentile shaded envelope) while the observations (AWAP over Australia and VCSN over New Zealand) are shown by the dashed line. The time series are given as anomalies relative to the mean of the entire period. The bias between the model and observations is indicated, along with the Pearson correlation coefficient (r) and p-value for testing non-correlation.





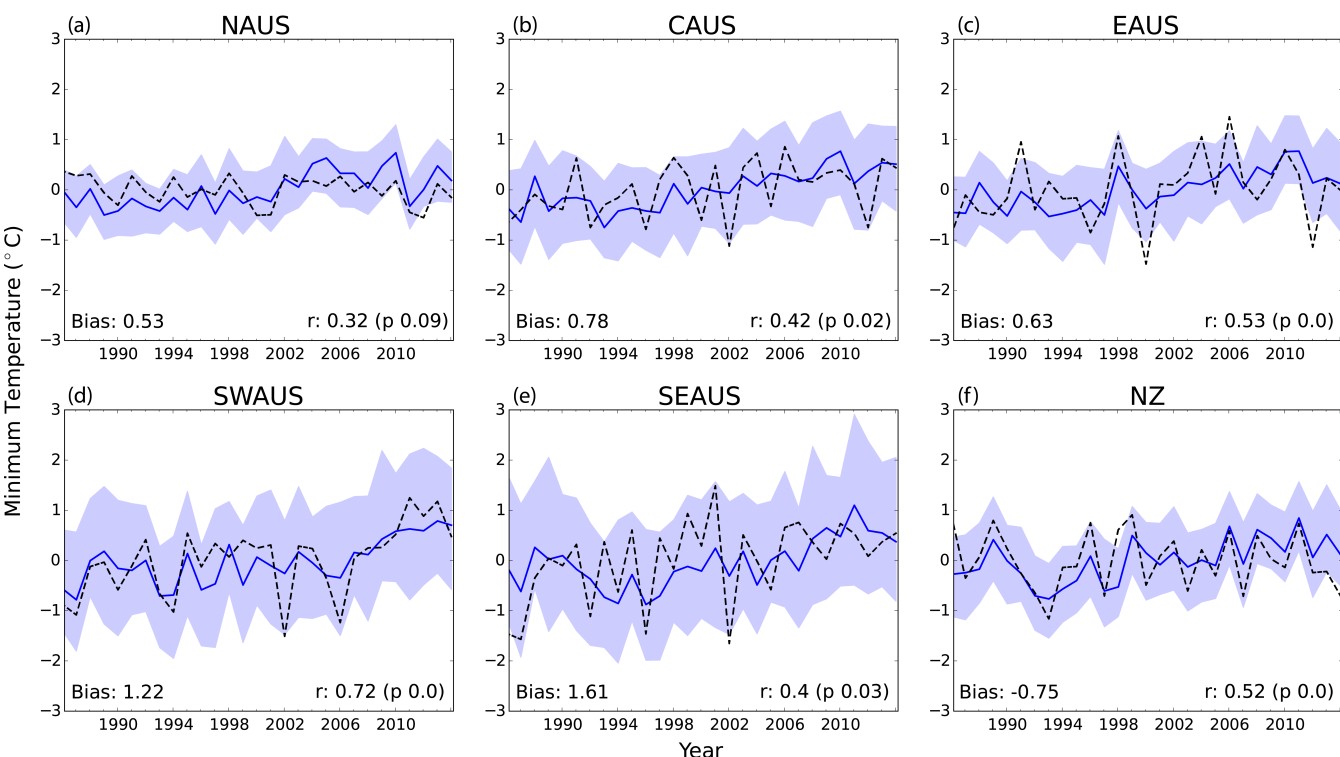

**Figure 6.** As in Figure 5, but showing average minimum temperature.





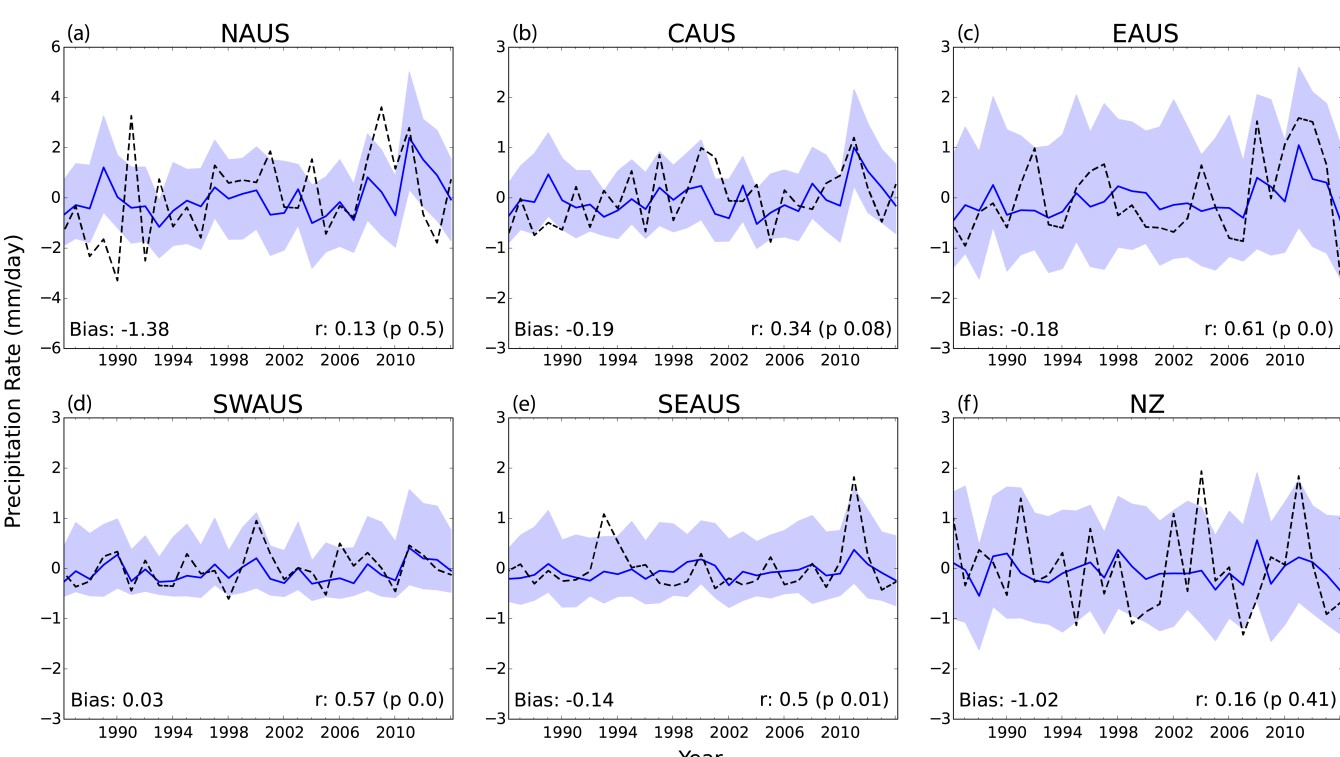

**Figure 7.** As in Figure 5, but showing average precipiation.





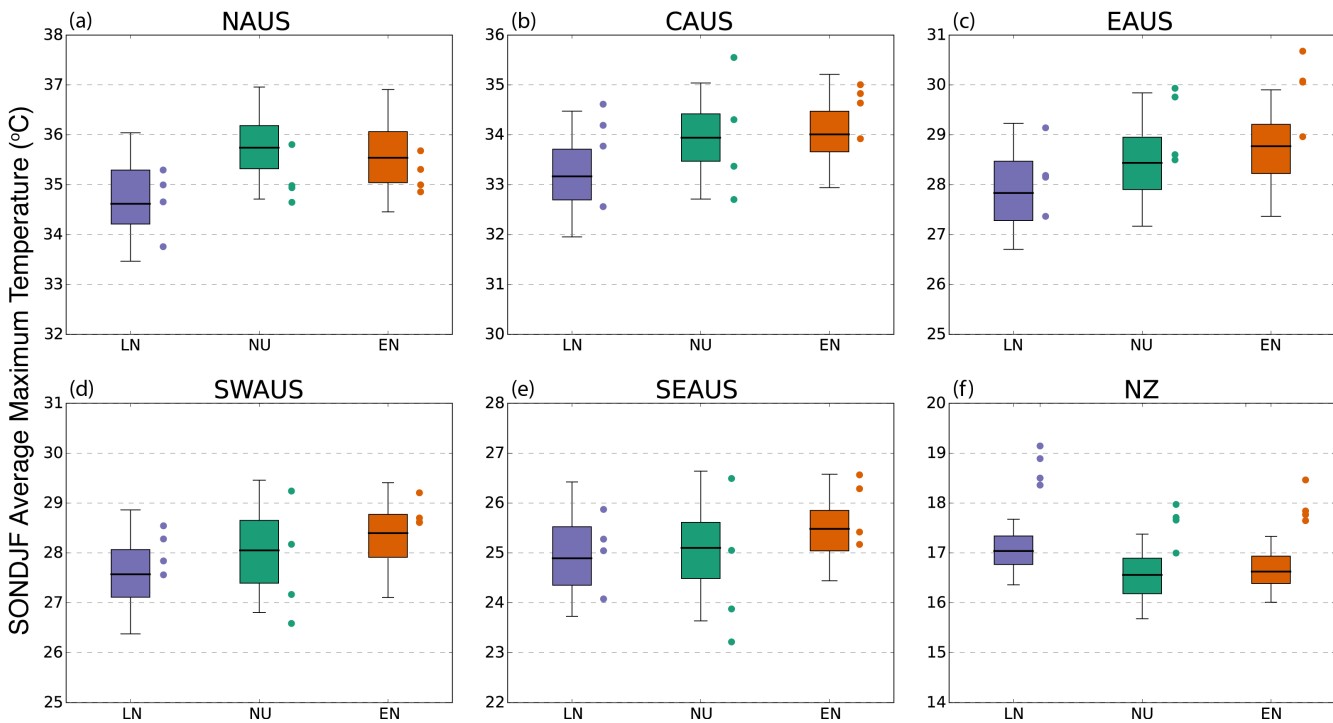

**Figure 8.** September–February average maximum temperature for the respective study regions (as labelled) during different phases of ENSO: La Niña (LN), neutral (NU) and El Niño (EN). Observed values are plotted as colored circles while values from the weather@home HadRM3P simulations are shown as box-and-whisker plots. The boxes show the median and interquartile range while the whiskers extend to the 5th and 95th percentiles. See text for details.





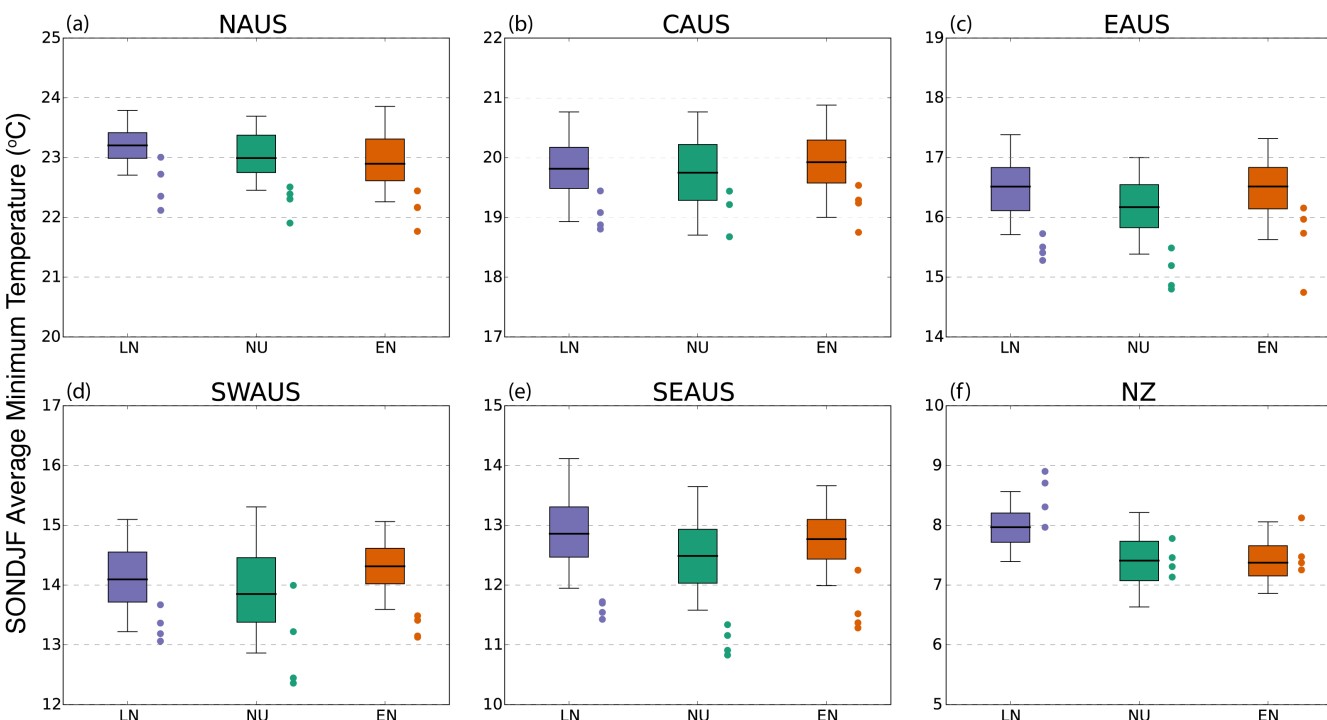

**Figure 9.** As in Figure 8, but showing minimum temperature.





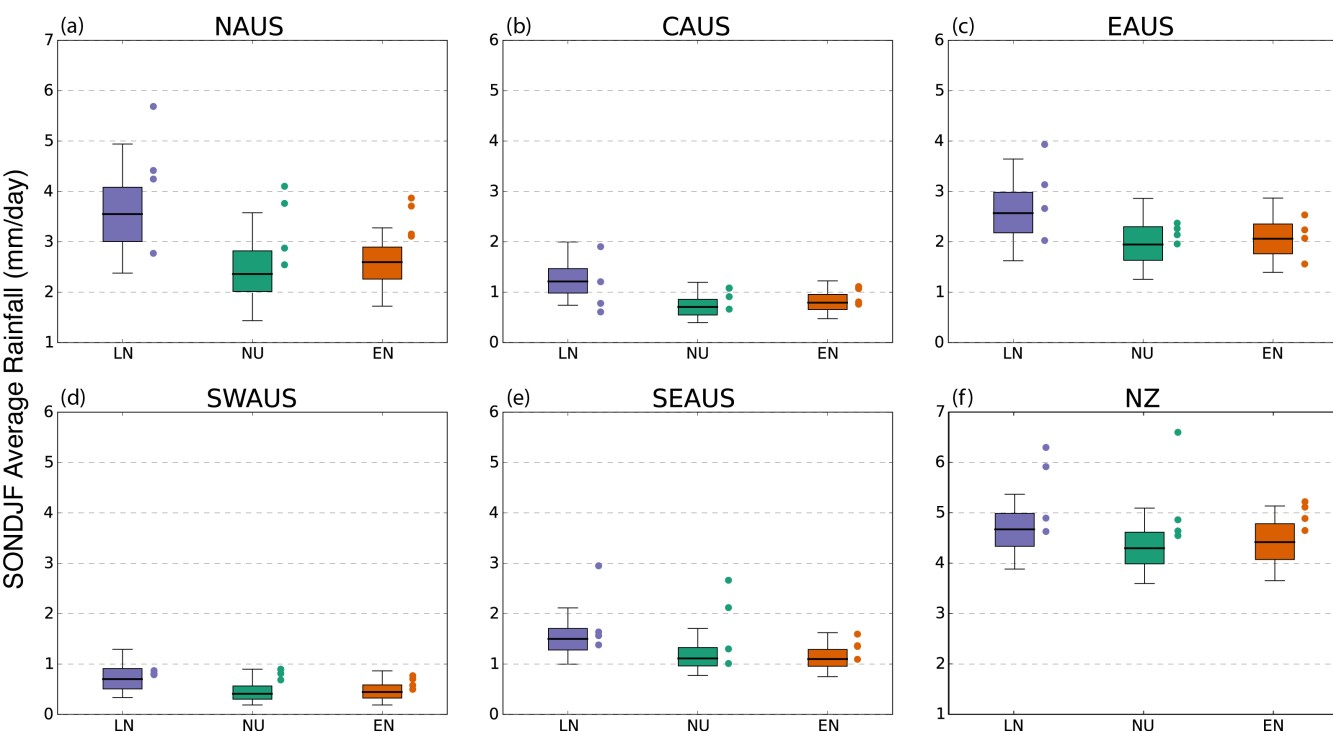

**Figure 10.** As in Figure 8, but showing precipitation.





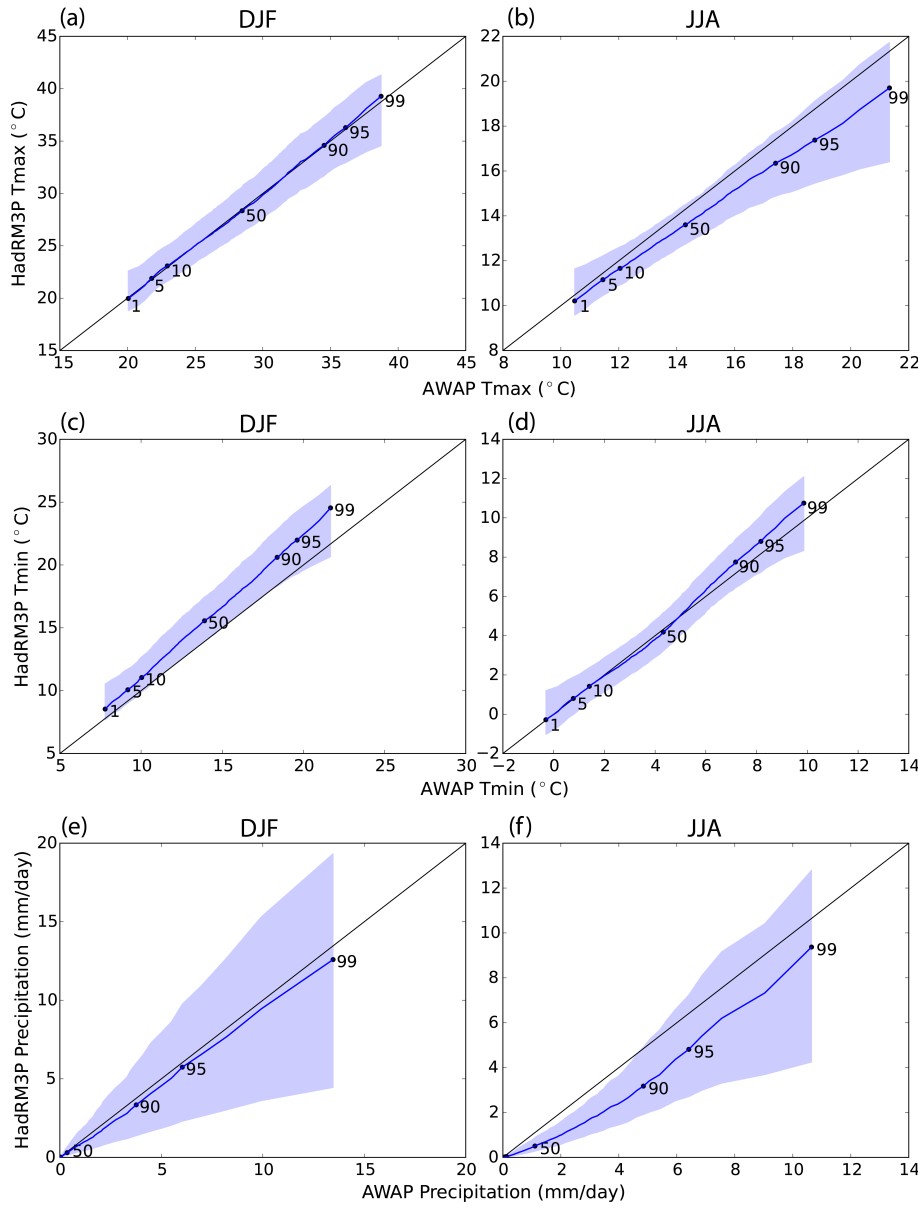

**Figure 11.** Quantile-quantile plots showing distributions of daily maximum temperature (a, b), minimum temperature (c, d) and precipitation (e, f), averaged over southeast Australia. Distributions are shown for December–February (a,c,e) and June–August (b, d, f). The solid blue line shows the percentile values for the entire ensemble of model simulations, while the blue envelope shows the 5th to 95th percentile range of values for individual ensemble members.





**Figure 12.** Estimated sea surface temperature response pattern (°C) to anthropogenic forcing, calculated from 10 different CMIP5 models (as labelled). The temperature responses are calculated for each month (January–December) but are shown here as annual averages.





**Figure 13.** Return periods of daily December–February maximum temperature at Melbourne, Australia, for observed climate conditions (red) and various counterfactual climate conditions (grey).