# Peer review of "The weather@home regional climate modelling project for Australia and New Zealand"

_Geoscientific Model Development, 2016_

## Referee Comment (RC1) · D. Stone (Referee) · 13 Jun 2016

This paper describes the experimental setup of an atmospheric modelling system for examination of extreme weather over the land territories of Australia and New Zealand in the context of anthropogenic climate change. It is well designed, well described in this paper, and various aspects of the output of the modelling system are adequately summarised. I recommend the paper for publication. I have some minor comments and suggested edits below, but I do not consider any of them to be required.

General:

You examine DJF and JJA values, and some SONDJF values. The onset/cessation seasons for temperature and (I think) rainfall occur during the SON and MAM seasons, and I believe extreme early/late onsets/cessations can be at least as important e.g. for

water resources and agriculture. Have you done these analyses for those seasons and are you able to summarise them? It probably does not have to be in the sort of detail done for DJF and JJA, but could just highlight any cases where e.g. the model might happen to be rather late (as proxied by the mean during the onset).

Technical:

page 1, lines 6-7 "more robust estimates of uncertainty" than what? You are using a single modelling system, so I am not sure how you can e.g. robustly estimate the uncertainty due to approximations in model design.

page 1, lines 12-13 Where or how? I do not see the URL of any data portal listed in the document, for instance.

page 1, line 16 I suppose wildfire might be considered a "climate-related" process, but I usually think of it as an ecological process.

page 1, lines 16-20 Are there any reports that document such "loss" and "tasks"?

page 2, line 1 Reisinger et alii (2014) (IPCC AR5 WGII Ch25) might be an appropriate reference here, as it is in this chapter of the IPCC AR5 that trends in Australian/New Zealand climate are assessed specifically.

page 2, line 10 "Distinguishing between internal" -> "Distinguishing between the responses to internal"

page 2, line 12 While the magnitude of the response is more or less the same at regional and global scales?

page 2, lines 20-22 Are these "drivers" or mechanisms? For instance, atmospheric blocking over Australia is a manifestation of climate variability over Australia, not a driver thereof.

page 2, line 31 "respond to anthropogenic" -> "respond to the absence of anthropogenic" This discussion concerns estimation of the counterfactual climate, correct?

page 6, line 9 Might the observations have any deficiencies?

page 6, lines 10-11 How do DJF and JJA project onto the wet, dry, onset, and/or cessation seasons over these regions?

page 8, lines 3-4 This is not clear to me for Tmin. The uncertainty on the median should be $\sim$sqrt(75)$\sim$8.7 times smaller than the range of the whiskers: are the respect Tmin median values larger than this? I cannot tell from the plot.

page 8, line 27 Maybe a 360-day calendar?

page 8, lines 32-33 How is "model uncertainty" estimated with just the single model?

page 9, lines 4-5 I am not sure, the observations lie within the spread of the simulations.

page 9, lines 9-10 The observations lie within the spread of the simulations for 11d, and are pretty much bang on for 11e.

page 9, lines 30-31 Re the sea ice extent, does this make sense for the Antarctic, where the recent trend has been toward slightly larger extent?

page 9, lines 32-33 You did not mention when describing the simulations if anything is done concerning land use/cover change. Is this included and, if so, how is this treated in the counterfactual simulations?

page 10, line 18 "quantified" -> "characterised" I do not see any reason to believe that these 10 estimates can be assumed to be uniformly sampled from a plausible posterior distribution. Rather than producing a posterior distribution, I think you are testing robustness against plausible alternative estimates.

page 10, lines 29-30 Where and/or how?

page 11, line 3 "allows extreme events" -> "allows certain types of extreme weather events"

---

## Short Comment (SC1) · 16 Jun 2016

General comments

This well written paper introduces a unique and valuable resource for event attribution over the Australia and New Zealand region, building on the success of equivalent systems focussing on different regions.

With regards reproducibility I have found some omissions in the model and experimental description that can be easily addressed. The system description is otherwise clear and well motivated.

A decent attempt to explore model uncertainty is made through the construction of 10 estimates of the counterfactual boundary conditions while the limitations of the system

are also discussed. The basic validation conducted is probably sufficient given that the studies that are performed with such a system generally require bespoke validation and are in fact already in print [Herring et al., 2015].

Altogether I found no issues that I consider major and so I would recommend this work for publication.

Specific comments

[p2] Use of phrase "internal ... climate forcings" While it is not unreasonable to refer to major modes of variability as forcing a regional climate akin to an external forcings it would be better to use a phrase like "internal climate variability".

[p2.7, also p2.20,23] "chaotic natural variability" is referring to variability generated internal to the climate system while in attribution we normally retain the phrase "natural variability" to refer to externally forced variability (i.e. by solar and volcanic forcing). "chaotic internal variability" would be better and is probably adequately distinguished from the major modes of internally generated variability in the context of this sentence. Later [p2.30] "natural forcing" is used in the normal sense so there is clear room for confusion.

Page 2.19-24 emphasises the importance of several major modes of internal variability in addition to ENSO (SAM, position of storm tracks, blocking) which are important in the Australia / New Zealand region but only ENSO is addressed in the remainder of the paper. I am not asking that additional validation be done for these other factors but could the authors comment on the relative importance of ENSO with respect to these others? For e.g. some reference to the literature to give an idea of the share of variance explained in inter-seasonal or monthly regional means of the two variables presented? It appears Risbey et al., 2009 Figs. 15 & 16 (already cited by the authors) may contain enough to go on.

[p2.28, also p9.22, Fig. 13 caption] Use of "observed" climate in reference to historical

climate. In a modelling context we would avoid using "observed" to refer to experiments with anthropogenic and natural forcings present as these are simply not observations but simulations whose climate is intended to reproduce an observed climate, but which may not do so. I advise substituting this with "anthropogenically forced", "historical" or similar. Elsewhere [p5.3] "historical ... climate scenario" is used.

[p4.33] Further initial condition perturbations are applied by selection of a range of start conditions with difference large scale circulation and soil moisture patterns. No details or references are given for the selection criteria or pattern generation. Please elaborate. Are the patterns physical / taken from a control?

[p5.4] Lower boundary conditions are taken from a daily analysis product (OSTIA) while previous versions of the weather@home experimental setup [Massey at al., 2015, Mote et al., 2015] used interpolation from a monthly observational dataset (HadISST1). Was there a good reason for this new choice? For instance was it felt that the daily analysis provides more faithful sub-monthly variability than interpolation from monthly data?

[5.10] Halocarbon prescription: experimental progenitor [Massey et al. 2015] prescribe a single halocarbon value designed to give the radiative forcing corresponding to the presence of all (6 AR4 recommended) represented species. Is the same manner of prescription used in these experiments or are the AR5 individual species concentrations separately prescribed?

[5.11] GHG concentrations and aerosol emissions. Could you be explicit about the concentrations that are prescribed post-2005? Specifically, is one of the RCP scenarios followed (for e.g. from http://www.pik-potsdam.de/∼mmalte/rcps/index.htm )?

Section 2 does not mention if land use changes are prescribed. I can see (from Massey et al., 2015, section 2.2.4) that fractions of surface types are specified. Is this specification fixed or time dependent? Are the prescribed fraction of the natural simulations representative of preindustrial conditions?

Section 2 I could not gather from this or Massey et al., 2015 what land surface scheme the models use.

Section 2 Spin down. Is there a spin down period allowed prior to creation of the experimental initial conditions or to the analysis period? Has any continued drift in climate variables such as soil moisture been seen over the 29 year historical experiment?

[p6.9] After taking 29 year means "any differences between the obs. and model output may be interpreted as model deficiencies". This is not strictly true as, for example, even the means of output from two members of the same model will still be subject to "standard error" which will decrease as 1/SQRT(n) for n data points. In comparing the difference of two such means the errors will also add in quadrature. Nevertheless I would estimate that the discussed biases depicted in the figures 2 – 4 easily stand out from this level of noise. Also I acknowledge that p6.15 says that this can be regarded as "an indication of model bias".

[p7.16] A comparison of time series variances, power spectra or quantile plots would provide a more objective measure of agreement than simply eye-balling that the obs. time series sits mostly inside the envelope of the models, especially given that the objective measure provided (correlation coefficients) will not allow an assessment of overall amplitude of the series. However given the intended brevity of the validation and later closer focus on daily data we can probably make do with this.

[p7.18] Precisely what series are the correlation coefficients between? Is it obs. and model median? This should go into caption to Fig. 5 also.

[Figures 5 - 7] Can you confirm that the p-values are for a one-sided test?

[p7.21] ENSO as driver of "natural climate variability" would better be "internal climate variability", again avoiding confusion with solar and volcanically forced variability.

[p9.30] What year or period are the "pre-industrial" GHG, ozone & aerosol levels taken from? Are these also CMIP5 recommended values?

[Figure]

[p10.1] "cannot be known" could better be phrased as "cannot be observed", which is indisputable. We may dispute whether the counterfactual world is knowable.

[p11.7-10] Use of an atmosphere only model is here portrayed as a limitation but the atmosphere only approach simply allows us to frame a different event attribution question than that provided by coupled experiments. Namely, we ask for the likelihood of an event subject to the lower boundary forcing provided by the precise phases of the various modes of oceanic (and cryospheric) variability at the time of the event, which is not possible with a coupled model.

Technical corrections

[p6.12] Could insert the word "daily" to be completely clear (sentence could be interpreted as maximum of seasonal average over 29 years, 75 members.). Ditto the caption to Figure 2.

[p7.14] "individual years" would better be phrased as "specific years".

[p9.26] Insert word "lower" before "boundary conditions" to distinguish from lateral.

[p9.32] "boundary conditions common to both" is incorrect here, should be "forcings common to both".

[p11.7] Unnecessary comma after "weather@home"

[Fig S1] "summertime" should be "wintertime" if genuinely June – August.

[p14.19 and elsewhere] Massey et al. "2014" should be "2015".

References

Herring, S. C., M. P. Hoerling, .J. P. Kossin, T. C. Peterson, and P. A. Stott, Eds., 2015: Explaining Extreme Events of 2014 from a Climate Perspective. Bull. Amer. Meteor. Soc., 96 (12), S1-S172.

Massey, N., et al. "weather@ home—development and validation of a very large

ensemble modelling system for probabilistic event attribution." Quarterly Journal of the Royal Meteorological Society 141.690 (2015): 1528-1545.

Mote, Philip W., et al. "Superensemble regional climate modeling for the western US." Bulletin of the American Meteorological Society 2015 (2015).

Risbey, James S., et al. "On the remote drivers of rainfall variability in Australia." Monthly Weather Review 137.10 (2009): 3233-3253.

---

## Author Comment (AC1) · 4 Aug 2016

Dear Dr Williams and readers,

Please find attached our revised manuscript with changes from the original version highlighted in red (see revised_manuscript.zip). We found the reviewers' comments to be very helpful and have responded to these comments below.

——-

Reviewer 1: Dáithí Stone

Reviewer's summary: This paper describes the experimental setup of an atmospheric modelling system for examination of extreme weather over the land territories of Australia and New Zealand in the context of anthropogenic climate change. It is well de-

signed, well described in this paper, and various aspects of the output of the modelling system are adequately summarised. I recommend the paper for publication. I have some minor comments and suggested edits below, but I do not consider any of them to be required.

1. Reviewer's comment (general): You examine DJF and JJA values, and some SONDJF values. The onset/cessation seasons for temperature and (I think) rainfall occur during the SON and MAM seasons, and I believe extreme early/late onsets/cessations can be at least as important e.g. for water resources and agriculture. Have you done these analyses for those seasons and are you able to summarise them? It probably does not have to be in the sort of detail done for DJF and JJA, but could just highlight any cases where e.g. the model might happen to be rather late (as proxied by the mean during the onset).

Authors' response: This additional analysis has been completed and is included within the supplementary material (see new Figures S11–S13). Overall, the weather@home model is seen to adequately resolve the onset/cessation seasons for temperature and precipitation.

2. Reviewer's comment (technical): page 1, lines 6-7 "more robust estimates of uncertainty" than what? You are using a single modelling system, so I am not sure how you can e.g. robustly estimate the uncertainty due to approximations in model design.

Authors' response: The manuscript has been revised to address this point (see page 1, line 7).

3. Reviewer's comment (technical): page 1, lines 12-13. Where or how? I do not see the URL of any data portal listed in the document, for instance.

Authors' response: The relevant information for accessing the data has been added under the section Data and code availability. Members of the research community wishing to access the data are encouraged to contact the authors directly. An online

data portal (with public URL) will become available in the future.

4. Reviewer's comment (technical): page 1, line 16. I suppose wildfire might be considered a "climate-related" process, but I usually think of it as an ecological process.

Authors' response: The lead author (Mitchell Black) is currently completing a PhD thesis that is using the weather@home ANZ framework to investigate the influence of anthropogenic climate change on wildfire risk. This material will be published in due course.

5. Reviewer's comment (technical): page 1, lines 16-20. Are there any reports that document such "loss" and "tasks"?

Authors' response: Citations have been included within the revised manuscript (see page 1 of revised manuscript, lines 17-19).

6. Reviewer's comment (technical): page 2, line 1. Reisinger et alii (2014) (IPCC AR5 WGII Ch25) might be an appropriate reference here, as it is in this chapter of the IPCC AR5 that trends in Australian/New Zealand climate are assessed specifically

Authors' response: This citation has been included within the revised manuscript (see page 2, line 3).

7. Reviewer's comment (technical): page 2, line 10 "Distinguishing between internal" -> "Distinguishing between the responses to internal"

Authors' response: Text revised accordingly (see page 2, line 12)

8. Reviewer's comment (technical): page 2, line 12. While the magnitude of the response is more or less the same at regional and global scales?

Authors' response: This sentence has been revised to allow the message to be clearer. (see page 2, lines 13-16).

9. Reviewer's comment (technical): page 2, lines 20-22 Are these "drivers" or mechanisms? For instance, atmospheric blocking over Australia is a manifestation of climate variability over Australia, not a driver thereof.

Authors' response: This sentence has been revised to remove this confusion between drivers and mechanisms (see page 2, lines 23-27).

10. Reviewer's comment (technical): page 2, line 31 "respond to anthropogenic" -> "respond to the absence of anthropogenic" This discussion concerns estimation of the counterfactual climate, correct?

Authors' response: This sentence has been revised accordingly (see page 2, line 35).

11. Reviewer's comment (technical): page 6, line 9 Might the observations have any deficiencies?

Authors' response: A sentence has been added to section Observational datasets (page 6, lines 11-13) to acknowledge potential limitations of the observational datasets used in this study. However, as indicated in the revised text, these datasets are nevertheless well regarded for use in model evaluation studies.

13. Reviewer's comment (technical): page 6, lines 10-11 How do DJF and JJA project onto the wet, dry, onset, and/or cessation seasons over these regions?

Authors' response: As indicated for point 1 above, additional analysis has been undertaken to examine the model's ability to resolve the onset/cessation seasons for temperature and precipitation. Overall, the model appears to adequately resolve this timing for each region. The supplementary material has been updated to include these additional figures and a brief summary of these results are included in the main body of manuscript. (see page 6, lines 22-23).

14. Reviewer's comment (technical): page 8, lines 3-4 This is not clear to me for Tmin. The uncertainty on the median should be âĹijsqrt(75)âĹij8.7 times smaller than the range of the whiskers: are the respect Tmin median values larger than this? I cannot tell from the plot.

Authors' response: Figures 8-10 have been updated to make the box plots clearer.

15. Reviewer's comment (technical): page 8, line 27 Maybe a 360-day calendar?

Authors' response: Manuscript updated accordingly.

16. Reviewer's comment (technical): page 8, lines 32-33 How is "model uncertainty" estimated with just the single model?

Authors' response: This has been corrected to indicate 'sampling uncertainty', not 'model uncertainty'.

17. Reviewer's comment (technical): page 9, lines 4-5 I am not sure, the observations lie within the spread of the simulations. page 9, lines 9-10 The observations lie within the spread of the simulations for 11d, and are pretty much bang on for 11e.

Authors' response: Indeed, this section of text was incorrect as the observed values fell within the model spread. This section of text has been updated to remove these incorrect statements (see page 9, lines 17-22).

18. Reviewer's comment (technical): page 9, lines 30-31 Re the sea ice extent, does this make sense for the Antarctic, where the recent trend has been toward slightly larger extent?

Authors' response: This point was investigated during the beta-testing phase of the weather@home Australia-New Zealand experiment. During this testing counterfactual climates were simulated using: 1.) sea ice extent corresponding to the year of maximum sea ice extent in the Southern Hemisphere from the OSTIA records (1985–2014), and 2.) sea ice extent corresponding to the year of minimum sea ice extent in the Southern Hemisphere. While not shown here, the results of these experiments identified that the choice of counterfactual sea ice extent had negligible impact on the resulting climates of Australia and New Zealand. Therefore, the weather@home Australia-New Zealand experiments presented in this manuscript used the maximum observed sea ice extent as a proxy for the counterfactual sea ice extent, for both hemispheres.

This is consistent with the methodology of the existing weather@home projects – that is, weather@home Europe (Massey et al. 2015) and weather@home Pacific North-West (Mote et al. 2015).

19. Reviewer's comment (technical): page 9, lines 32-33 You did not mention when describing the simulations if anything is done concerning land use/cover change. Is this included and, if so, how is this treated in the counterfactual simulations?

Authors' response: The weather@home ANZ setup uses the MOSES1.0 land surface scheme. The surface type is fixed and is the same between the historical and counterfactual climate simulations. The manuscript has been revised to include this information (see page 4, paragraph 1).

20. Reviewer's comment (technical): page 10, line 18 "quantified" -> "characterised" I do not see any reason to believe that these 10 estimates can be assumed to be uniformly sampled from a plausible posterior distribution. Rather than producing a posterior distribution, I think you are testing robustness against plausible alternative estimates.

Authors' response: Manuscript updated accordingly.

21. Reviewer's comment (technical): page 10, lines 29-30 Where and/or how?

Authors' response: The relevant information for accessing the data has been added under section Data and code availability (page 12, paragraph 1).

22. Reviewer's comment (technical): "allows extreme events" -> "allows certain types of extreme weather events"

Authors' response: Manuscript updated accordingly.

References

Massey, N. et al (2015). Weather@home – development and validation of a very large ensemble modelling system for probabilistic event attribution. Quarterly Journal of the

Royal Meteorological Society, doi:10.1002/qj.2455

Mote, P. et al (2015). Superensemble regional climate modeling for the western US. Bulletin of the American Meteorological Society, doi: 10.1175/BAMS-D-14-00090.1

Please also note the supplement to this comment:
http://www.geosci-model-dev-discuss.net/gmd-2016-100/gmd-2016-100-AC1-supplement.zip

---

## Author Comment (AC2) · 4 Aug 2016

Dear Dr Williams and readers,

Please find attached our revised manuscript with changes from the original version highlighted in red (see revised_manuscript.zip uploaded as part of the author reply to RC1). We found the reviewers' comments to be very helpful and have responded to these comments below

———

Reviewer 2: Andrew Ciavarella

Reviewer's summary: This well written paper introduces a unique and valuable resource for event attribution over the Australia and New Zealand region, building on

the success of equivalent systems focussing on different regions. With regards reproducibility I have found some omissions in the model and experimental description that can be easily addressed. The system description is otherwise clear and well motivated. A decent attempt to explore model uncertainty is made through the construction of 10 estimates of the counterfactual boundary conditions while the limitations of the system are also discussed. The basic validation conducted is probably sufficient given that the studies that are performed with such a system generally require bespoke validation and are in fact already in print [Herring et al., 2015]. Altogether I found no issues that I consider major and so I would recommend this work for publication.

1. Reviewer's comment (specific): [p2] Use of phrase "internal ... climate forcings" While it is not unreasonable to refer to major modes of variability as forcing a regional climate akin to an external forcings it would be better to use a phrase like "internal climate variability".

Authors' response: Manuscript updated accordingly

2. Reviewer's comment (specific): [p2.7, also p2.20,23] "chaotic natural variability" is referring to variability generated internal to the climate system while in attribution we normally retain the phrase "natural variability" to refer to externally forced variability (i.e. by solar and volcanic forcing). "chaotic internal variability" would be better and is probably adequately distinguished from the major modes of internally generated variability in the context of this sentence. Later [p2.30] "natural forcing" is used in the normal sense so there is clear room for confusion.

Authors' response: Manuscript updated accordingly

3. Reviewer's comment (specific): Page 2.19-24 emphasises the importance of several major modes of internal variability in addition to ENSO (SAM, position of storm tracks, blocking) which are important in the Australia / New Zealand region but only ENSO is addressed in the remainder of the paper. I am not asking that additional validation be done for these other factors but could the authors comment on the relative importance

of ENSO with respect to these others? For e.g. some reference to the literature to give an idea of the share of variance explained in inter-seasonal or monthly regional means of the two variables presented? It appears Risbey et al., 2009 Figs. 15 & 16 (already cited by the authors) may contain enough to go on.

Authors' response: As Reviewer 1 correctly identified, the original body of text confused drivers of internal climate variability (e.g., ENSO) from manifestations of climate variability (e.g., blocking). The text was revised to address this point. As the text now stands, three drivers of climate variability are identified – ENSO, Southern Annular Mode and the Indian Ocean Dipole. While each of these three modes of variability have an important influence on Australian temperature and precipitation extremes (e.g., Risbey et al. 2009, Min et al. 2013), ENSO was examined in this manuscript as it is the mode of variability that has been examined in a number of event attribution studies to date (e.g., King et al. 2013; Lewis and Karoly 2013; Christidis et al. 2013) and there are a number of planned studies wanting to specifically investigate the role of ENSO (e.g., Black and Karoly, 2016; Karoly et al. 2016). For brevity, the analysis for IOD and SAM was not presented in the current manuscript but will feature in an upcoming publication. Analysis of blocking within the weather@home ANZ model is provided in the recent publication by Grose et al. (2015).

4. Reviewer's comment (specific): [p2.28, also p9.22, Fig. 13 caption] Use of "observed" climate in reference to historical climate. In a modelling context we would avoid using "observed" to refer to experiments with anthropogenic and natural forcings present as these are simply not observations but simulations whose climate is intended to reproduce an observed climate, but which may not do so. I advise substituting this with "anthropogenically forced", "historical" or similar. Elsewhere [p5.3] "historical ... climate scenario" is used.

Authors' response: The manuscript has been updated to replace 'observed' with 'historical'

5. Reviewer's comment (specific): [p4.33] Further initial condition perturbations are applied by a range of start conditions with difference large scale circulation and soil moisture patterns. No details or references are given for the selection criteria or pattern generation. Please elaborate. Are the patterns physical / taken from a control?

Authors' response: For the weather@home ANZ experiment, each model simulation is initialised using a restart file created in a control simulation for the previous year. For each experiment, 100 unique restart files are created (each with a different atmospheric state and soil moisture pattern/profile).

For example, consider the weather@home experiment for the year 2015. For this experiment, the model is run 100 times for the preceding year (2014) and the resulting restart files from these experiments are used to initialise the 2015 simulations. Therefore, there are essentially 100 'groups' of simulations created for the year 2015, with members of each 'group' containing the same restart file. For members within each 'group', slightly different initial condition perturbations are applied to the three-dimensional potential profile of the restart file (as described within the manuscript).

The manuscript has revised to clarify this (see page 5).

6. Reviewer's comment (specific): [p5.4] Lower boundary conditions are taken from a daily analysis product (OSTIA) while previous versions of the weather@home experimental setup [Massey at al., 2015, Mote et al., 2015] used interpolation from a monthly observational dataset (HadISST1). Was there a good reason for this new choice? For instance was it felt that the daily analysis provides more faithful sub-monthly variability than interpolation from monthly data?

Authors' response: All recent weather@home experiments have transitioned to using sea surface temperatures and sea ice extent from the OSTIA dataset (e.g., see Schaller et al. 2016, Mitchell et al. 2016). As the reviewer correctly identified, these daily fields provide a more faithful estimate of sub-monthly variability than is achieved from interpolation of monthly data.

**[GMDD]{.orange}**

Interactive
comment

7. Reviewer's comment (specific): [5.10] Halocarbon prescription: experimental pro-genitor [Massey et al. 2015] prescribe a single halocarbon value designed to give the radiative forcing corresponding to the presence of all (6 AR4 recommended) repre-sented species. Is the same manner of prescription used in these experiments or are the AR5 individual species concentrations separately prescribed?

Authors' response: As per Massey et al. (2015), a single halocarbon value is used to give the radiative forcing corresponding to the presence of all represented species. The manuscript has been updated to include this information (see page 5, paragraph 2).

8. Reviewer's comment (specific): [5.11] GHG concentrations and aerosol emis-sions. Could you be explicit about the concentrations that are prescribed post-2005? Specifically, is one of the RCP scenarios followed (for e.g. from http://www.pik-potsdam.de/âĹijmmalte/rcps/index.htm )?

Authors' response: Post-2005, the greenhouse gas concentrations and aerosol emis-sions follow the RCP 8.5 scenario. The manuscript has been updated to include this information (see page 5, paragraph 2).

9. Reviewer's comment (specific): Section 2 does not mention if land use changes are prescribed. I can see (from Massey et al., 2015, section 2.2.4) that fractions of surface types are specified. Is this specification fixed or time dependent? Are the prescribed fraction of the natural simulations representative of preindustrial conditions? Section 2 I could not gather from this or Massey et al., 2015 what land surface scheme the models use.

Authors' response: The model uses the MOSES 1.0 land surface scheme with fixed surface type. There is not change in surface type between the historical and counter-factual climate scenarios. The manuscript has been updated to include this information (see page 4, paragraph 1).

[Printer-friendly version]{.orange}

[Discussion paper]{.orange}

10. Reviewer's comment (specific): Section 2 Spin down. Is there a spin down period allowed prior to creation of the experimental initial conditions or to the analysis period? Has any continued drift in climate variables such as soil moisture been seen over the 29 year historical experiment?

Authors' response: As indicated above, each model year is initialised from a restart file output from a model simulation for the previous year. Therefore, there is a 12-month spin-down period allowed. Continuous integration of the model over an extended period (1985–2014) has not revealed any continued drift in soil moisture.

11. Reviewer's comment (specific): [p6.9] After taking 29 year means "any differences between the obs. and model output may be interpreted as model deficiencies". This is not strictly true as, for example, even the means of output from two members of the same model will still be subject to "standard error" which will decrease as 1/SQRT(n) for n data points. In comparing the difference of two such means the errors will also add in quadrature. Nevertheless I would estimate that the discussed biases depicted in the figures 2 – 4 easily stand out from this level of noise. Also I acknowledge that p6.15 says that this can be regarded as "an indication of model bias".

Authors' response: As the reviewer indicates, the biases depicted in Figures 2–4 easily stand out from sampling noise and can be regarded as an indication of model bias. The purpose of these figures is to provide a general overview of model performance; we feel that these images are able to portray this required information in a simple and adequate manner.

12. Reviewer's comment (specific): [p7.16] A comparison of time series variances, power spectra or quantile plots would provide a more objective measure of agreement than simply eye-balling that the obs. time series sits mostly inside the envelope of the models, especially given that the objective measure provided (correlation coefficients) will not allow an assessment of overall amplitude of the series. However given the intended brevity of the validation and later closer focus on daily data we can probably

make do with this.

Authors' response: We thank the author for these suggestions. Given the brevity of the paper we too feel that the level of information currently presented is appropriate.

13. Reviewer's comment (specific): [p7.18] Precisely what series are the correlation coefficients between? Is it obs. and model median? This should go into caption to Fig. 5 also.

Authors' response: The correlation coefficients are calculated between the observation and model medians. The manuscript has been updated to include this, including the caption of Figure 5.

14. Reviewer's comment (specific): [Figures 5 - 7] Can you confirm that the p-values are for a one-sided test?

Authors' response: The p-values are for a two-sided test. The figure captions have been updated to include this information.

15. Reviewer's comment (specific): [p7.21] ENSO as driver of "natural climate variability" would better be "internal climate variability", again avoiding confusion with solar and volcanically forced variability.

Authors' response: Manuscript updated accordingly.

16. Reviewer's comment (specific): [p9.30] What year or period are the "pre-industrial" GHG, ozone & aerosol levels taken from?

Authors' response: The manuscript has been updated to include this information.

17. Reviewer's comment (specific): [p10.1] "cannot be known" could better be phrased as "cannot be observed", which is indisputable. We may dispute whether the counter-factual world is knowable.

Authors' response: Manuscript updated accordingly.

[Figure]

18. Reviewer's comment (specific): [p11.7-10] Use of an atmosphere only model is here portrayed as a limitation but the atmosphere only approach simply allows us to frame a different event attribution question than that provided by coupled experiments. Namely, we ask for the likelihood of an event subject to the lower boundary forcing provided by the precise phases of the various modes of oceanic (and cryospheric) variability at the time of the event, which is not possible with a coupled model.

Authors' response: The reviewer has raised an important point – the atmosphere only model is not necessarily a limitation. The text has been revised accordingly (see page 11, paragraph 3).

19. Reviewer's comment (technical corrections): [p6.12] Could insert the word "daily" to be completely clear (sentence could be interpreted as maximum of seasonal average over 29 years, 75 members.). Ditto the caption to Figure 2.

Authors' response: Manuscript updated accordingly.

20. Reviewer's comment (technical corrections): [p7.14] "individual years" would better be phrased as "specific years".

Authors' response: Manuscript updated accordingly.

21. Reviewer's comment (technical corrections): [p9.26] Insert word "lower" before "boundary conditions" to distinguish from lateral.

Authors' response: Manuscript updated accordingly.

22. Reviewer's comment (technical corrections): [p9.32] "boundary conditions common to both" is incorrect here, should be "forcings common to both".

Authors' response: Manuscript updated accordingly.

23. Reviewer's comment (technical corrections): [p11.7] Unnecessary comma after "weather@home"

Authors' response: Manuscript updated accordingly.

24. Reviewer's comment (technical corrections): [Fig S1] "summertime" should be "wintertime" if genuinely June – August.

Authors' response: Manuscript updated accordingly.

25. Reviewer's comment (technical corrections): [p14.19 and elsewhere] Massey et al. "2014" should be "2015".

Authors' response: Manuscript updated accordingly.

References

Black, M.T. and Karoly, D.J. (2016). Climate change was an important driver of southern Australia's warmest October on record [in "Explaining Extreme Events of 2015 from a Climate Perspective"], Bulletin of the American Meteorological Society, under review.

Christidis, N et al. (2013). An attribution study of the heavy rainfall over eastern Australia in March 2012 [in "Explaining Extreme Events of 2012 from a Climate Perspective"], Bulletin of the American Meteorological Society, 94, S58-S61.

Grose, M. R. et al. (2015) Attribution of exceptional mean sea level pressure anomalies south of Australia in August 2014 [in "Explaining Extreme Events of 2014 from a Climate Perspective"], Bulletin of the American Meteorological Society, 96, S158–S162.

Karoly, D.J. et al. (2016). The roles of climate change and El Niño in the record low rainfall in October 2015 in Tasmania, Australia [in "Explaining Extreme Events of 2015 from a Climate Perspective"], Bulletin of the American Meteorological Society, under review.

King, A et al. (2013). Limited Evidence of Anthropogenic Influence on the 2011-12 Extreme Rainfall over Southeast Australia [in "Explaining Extreme Events of 2012 from a Climate Perspective"], Bulletin of the American Meteorological Society, 94, S55-S58.

Lewis, S and Karoly, D.J (2013). Anthropogenic contributions to Australia's record summer temperatures of 2013, Geophysical Research Letters, 40, 3705-3709.

Mitchell, D. et al. (2016). Attributing human mortality during extreme heat waves to anthropogenic climate change. Environmental Research Letters, doi: 10.1088/1748-9326/11/7/074006

Schaller, N. et al. (2016). The human influence on climate in the winter 2013/2014 floods in southern England. Nature Climate Change, doi: 10.1038/NCLIMATE2927
* * *